# CD137 Signaling Is Critical in Fungal Clearance during Systemic *Candida albicans* Infection

**DOI:** 10.3390/jof7050382

**Published:** 2021-05-14

**Authors:** Vuvi G. Tran, Na N. Z. Nguyen, Byungsuk Kwon

**Affiliations:** School of Biological Science, University of Ulsan, Ulsan 44610, Korea; nguyennuzenna@gmail.com

**Keywords:** *Candida albicans*, CD137, neutrophil, fungal clearance

## Abstract

Invasive fungal infections by *Candida albicans* frequently cause mortality in immunocompromised patients. Neutrophils are particularly important for fungal clearance during systemic *C. albican* infection, yet little has been known regarding which surface receptor controls neutrophils’ antifungal activities. CD137, which is encoded by *Tnfrsf9*, belongs to the tumor necrosis receptor superfamily and has been shown to regulate neutrophils in Gram-positive bacterial infection. Here, we used genetic and immunological tools to probe the involvement of neutrophil CD137 signaling in innate defense mechanisms against systemic *C. albicans* infection. We first found that *Tnfrsf9*^−/−^ mice were susceptible to *C. albicans* infection, whereas injection of anti-CD137 agonistic antibody protected the host from infection, suggesting that CD137 signaling is indispensable for innate immunity against *C. albicans* infection. Priming of isolated neutrophils with anti-CD137 antibody promoted their phagocytic and fungicidal activities through phospholipase C. In addition, injection of anti-CD137 antibody significantly augmented restriction of fungal growth in *Tnfrsf9*^−/−^ mice that received wild-type (WT) neutrophils. In conclusion, our results demonstrate that CD137 signaling contributes to defense mechanisms against systemic *C. albicans* infection by promoting rapid fungal clearance.

## 1. Introduction

Neutrophils represent a major group of effector cells that are critical for defense against invasive fungal infection [1,2]. They employ various strategies to restrict fungi, including the machinery for capturing and killing, such as various cytotoxic enzymes and their products, neutrophil extracellular traps (NETs), and phagocytic receptors [2]. In mouse, C-type lectin receptors such as Dectin-1, Dectin-2, and Dectin-3, and Ephrin type-A receptor 2 (Eph2A) can directly recognize the cell wall components of *C. albicans* in macrophages, dendritic cells, and neutrophils [3,4,5,6,7,8,9]. Complement receptor 3 (CR3) and Fc-gamma receptors (FcγRs) are strongly expressed in neutrophils and represent phagocytic receptors recognizing opsonized *C. albicans* [10,11,12]. Phagocytic receptors cooperate with each other to enhance phagocytosis: for example, Mayadas and colleagues have demonstrated that engagement of Dectin-1 with β-glucan, a cell wall component of *C. albicans*, enhances the phagocytic activity of neutrophils through CR3 activation [3]. We have further demonstrated that IL-33 enforces Dectin-1 signaling to increase the expression of CR3, whereby IL-33 effectively restricts *C. albicans* [4]. Recognition of opsonized *C. albicans* by EphA2 can also augment FcγR-mediated killing of extracellular yeast forms in such a way that the two signals are converged to synergistically enhance release of reactive oxygen species (ROS) [9]. Cooperation of these fungi-recognition receptors, NETs and neutrophil swarming is particularly effective in inhibiting clustering of *Candida* hyphae and thereby preventing their outgrowth within organs [13]. These experimental outcomes may make it possible to design therapies that could selectively enhance neutrophil function in the context of fungal clearance. Accordingly, manipulation of neutrophil phagocytic and fungicidal function is considered to be an attractive avenue that could benefit patients during fungal infections.

CD137 is well known for its costimulatory function for CD8^+^ T cells [14]. However, its expression is detected in a broad range of cells, which has implications that CD137 could function in a variety of cells, including non-T-cell lymphoid cells, myeloid cells, and non-hematopoietic cells. For example, there are reports showing that neutrophil-specific CD137 signaling plays an important role in defense against bacterial infections [15,16,17,18]. As yet, it remains to be clarified whether CD137 is involved in innate defense against invasive fungal infections. In this study, we specified CD137 function in neutrophils during systemic *C. albicans* infection. We show that anti-CD137 agonistic antibody enhances a neutrophil’s ability to restrict fungal growth.

## 2. Materials and Methods

### 2.1. Mice, Antibodies, and Reagents

C57BL/6 mice were purchased from Orient Bio-Charles River (Seongnam, Korea). *Tnfrsf9*^−/−^ mice were originally generated by Dr. Byoung S. Kwon (Eutilex, Seoul, Korea) [19], and they were backcrossed to C57BL/6 mice more than 15 generations. These mice are healthy and do not show any phenotypic abnormality. They were maintained in a specific pathogen-free facility and used when 7–8 weeks old. All experiments were conducted according to the regulations issued by the Animal Committee of the University of Ulsan. Anti-CD137 (3H3) and anti-Gr-1 (RB6-8C5) monoclonal antibodies were purified from ascites fluid. Control rat IgG was purchased from Millipore Sigma Korea (Seoul, Korea). Anti-CD16/CD32 (2.4G2) antibody was purified from hybridoma culture. The following FITC-, PE-, APC-, PerCP-, APC-cy7- or biotin-conjugated antibodies to mouse proteins were purchased from BD Biosciences or e-Bioscience: CD45 (30-F11), CD11b (M1/70), Ly6G (1A8), and Ly6C (AL-21).

### 2.2. Fungal Strains, Growth Conditions, and Infection

*C. albicans* (ATCC26555) was grown in peptone dextrose extract at 30 °C overnight, and aliquots were frozen at −80 °C. To kill the *C. albicans* yeasts, the organisms were harvested by centrifugation, and pellets were washed twice in sterile PBS. After resuspension at a density of 1 × 10^8^ cells/mL, heat killing was performed at 90 °C for 30 min. To induce experimental candidiasis, *C. albicans* was intravenously inoculated into the lateral caudal tail vein at a dose of 3 × 10^5^ colony-forming units (CFUs). For counting CFUs, mice were euthanized and kidneys were removed aseptically. Harvested kidneys were homogenized in 2 mL of PBS, and serial dilutions of homogenates were plated on Sabouraud agar and incubated at 37 °C for 24 h. Colonies were counted, and results were expressed as log_10_(CFUs/mL) or log_10_(CFUs/organ).

### 2.3. Isolation of Neutrophils

Total bone marrow cells were collected from tibias and femurs of 8-week old mice by flushing with RPMI 1640 media, filtered through a sterile mesh (Corning, Glendale, AZ, USA), and washed. After erythrocytes were lysed in hemolysis buffer (144 mM NH_4_Cl and 17 mM Tris-HCl [pH 7.2]), the remaining cells were resuspended in MACS buffer (1x PBS containing EDTA and 3% calf serum). Neutrophils were isolated using anti-Ly6G MACS microbeads, according to manufacturer’s protocols (Miltenyi Biotech Korea, Seoul, Korea). The purity of neutrophils routinely reached >98%.

### 2.4. Depletion of Neutrophils

Neutrophil depletion was achieved by intraperitoneally injecting 200 μg of anti-Gr-1 (RB6-8C5) monoclonal antibody into mice 2 days before *C. albicans infection*. To determine the extent of depletion, blood was harvested 1 day after antibody injection, erythrocytes were lysed in hemolysis buffer, and the remaining cells were used for staining with anti-Ly6G antibody. More than 95% neutrophils were routinely depleted by anti-Gr-1 antibody [4,19,20,21,22].

### 2.5. Adoptive Transfer of Neutrophils

For adoptive transfer, neutrophils were purified from WT mice, as described in Section 2.3. Five millions of neutrophils per mouse were intravenously injected into *Tnfrsf9*^−/−^ mice and then they received control or anti-CD137 antibody (200 μg per mouse) immediately after adoptive transfer. Six hours later, mice were infected with *C. albicans*.

### 2.6. Preparation of Kidney Cells

Kidneys were perfused, minced, and placed in DMEM (Gibco) containing 1 μg/mL collagenase IA (Millipore Sigma Korea, Seoul, Korea) at 37 °C for 30 min. Digested kidney tissues were passed through a 40-µm cell strainer (Corning, Glendale, AZ, USA), and the cell suspensions obtained were centrifuged at 300× *g* for 10 min. Cells were then washed in PBS containing 2% BSA, were suspended in 36% Percoll (GE Healthcare, Chalfont, UK), and were gently overlaid onto 72% Percoll. After centrifugation at 900× *g* for 30 min at room temperature, cells were retrieved from the Percoll interface and washed twice in DMEM medium and once with staining buffer (PBS containing 2% BSA and 0.1% sodium azide).

### 2.7. Flow Cytometry

Prepared cells were blocked with 2.4G2 monoclonal antibody in staining buffer (PBS containing 0.2% BSA and 0.1% sodium azide) at 4 °C for 20 min, incubated with relevant mAbs at 4 °C for 30 min, and then rewashed twice with staining buffer. Flow cytometric analysis was performed using a FACS Canto II unit (BD Biosciences), and the data were analyzed using FACS Diva (BD Biosciences, San Jose, CA, USA) and FlowJo software (Tree Star, Ashland, OR, USA).

### 2.8. Analysis of Renal Function

To determine kidney functions, concentrations of creatinine and blood urea nitrogen (BUN) in sera were measured colorimetrically using the Quantichrom Urea Assay and the Quantichrom Creatine Assay kits (Bioassay Systems, Hayward, CA, USA). For measurement of BUN, 5 μL of serum samples in duplicate were transferred into wells of a clear bottom 96-well plate. Reaction was done by adding 200 μL of working reagent (1:1 mix of Reagent A and Reagent B) and by incubating for 20 min. Optical density at 520 nm was read and urea concentration was calculated from the standard curve. Measurement of creatinine was done in a similar way, but optical density was read at 510 nm immediately, and then at 5 min after reaction.

### 2.9. Pathological Scoring

Kidneys were fixed in 10% (vol/vol) formalin, paraffin-embedded, sectioned (5 μm), stained with hematoxylin and eosin (H&E) or Periodic acid-Schiff (PAS), and analyzed. Kidney injury was scored by a single pathologist as the percentage of damaged tubules in the corticomedullary junction. Criteria for kidney injury included tubular necrosis, cast formation, loss of brush border, tubular dilatation, and immune cell infiltration. Scoring for each category was as follows: 0, no change; 1, <10%; 2, 11–25%; 3, 25–45% area change. Scores for all the categories were added for the final injury scoring.

### 2.10. Measurement of Cytokines and Chemokines

Cytokines and chemokines present in total kidney homogenates were measured by ELISA (ThermoFisher Scientitic Korea, Seoul), according to the manufacturer’s protocols. In brief, ELISA plate was incubated with a capture antibody in a coating buffer overnight at 4 °C. After washing 3 times with 250 μL of Washing Buffer, wells were blocked with 200 μL of 1× Assay Diluent at room temperature for 1 h. One hundred μL of samples was incubated at room temperature for 2 h, followed by addition of detection antibody diluted in 1× Assay Diluent and incubation for 1 h. After thorough washing, wells were added by 100 μL of Avidin-HRP diluted in 1× Assay Diluent. Reaction was done by adding 100 μL of substrate for 15 min and then stopped by adding 5 μL of Stop Solution. Plate was read at 450 nm.

### 2.11. Phagocytosis Assay

In vivo phagocytosis assays were performed as described previously [4]. In brief, mice were intraperitoneally injected with anti-CD137 antibody (200 μg per mouse) 1 h before infusion of 5 × 10^8^ CFU FITC-labeled heat-killed (HK) *C. albicans*. One-hour later, peritoneal cells were harvested and stained with anti-CD11b and anti-Ly6G monoclonal antibodies on ice. Percentages of neutrophils containing phagocytosed *C. albicans* were determined by flow cytometry. In vitro phagocytosis assays were performed as previously described [4,20,22]. In brief, neutrophils were purified from the bone marrow using anti-Ly6G microbeads. HK *C. albicans* was labeled with FITC and opsonized, and then added to neutrophils at 37 °C for 20 min (multiplicity of infection [MOI] = 10). Phagocytosis was stopped by transferring of cells into ice, and cells were then washed thoroughly with cold FACS buffer. Extracellular fluorescence was quenched by adding 200 µL of PBS containing 0.04% trypan blue and 1% formaldehyde. Cells containing fungi were counted by flow cytometry. Phagocytosis (%) was expressed as the percentage of neutrophils that phagocytosed FITC-labeled *C. albicans*. The mean fluorescent intensity (MFI) was also obtained to see the extent of phagocytosis per cell.

### 2.12. Fungicidal Assay

Live *C. albicans* was opsonized with mouse serum and added to neutrophils (MOI = 10). The mixture was incubated at 37 °C with shaking for 20 min to allow the phagocytosis of live *C. albicans*. Cells were then washed thoroughly in cold PBS, were resuspended in warm DMEM medium, and were further incubated at 37 °C. At indicated times, a 200-mL sample was taken, cells were lysed in PBS containing 0.1% Triton X-100, and CFUs were enumerated by plating on agar. The killing (%) was calculated as [1 − (CFUs after incubation/phagocytized CFUs at the start of incubation)] × 100.

### 2.13. ROS Production

ROS were detected using the fluorescent probe DCF-DA (Invitrogen). Isolate neutrophils (3 × 10^5^) were seeded into 96-well plate and primed with anti-CD137 (5 μg/mL) for 2 h prior to being challenged by opsonized HK *C. albicans* (MOI = 10). After 30-min incubation, DCF-DA (2 μM) was added and fluorescence was measured by a fluorescent plate reader at an excitation/emission = 485/530 nm at an interval of 10 min for 1 h. The fluorescence intensity was defined as the relative fluorescence units.

### 2.14. Statistical Analysis

All data were analyzed using GraphPad Prism5 (GraphPad Software, San Diego, CA, USA). Survivals and unpaired data were analyzed using the log rank test and the *t*-test, respectively. Results are expressed as means ± SEMs. Statistical significance was accepted for *p* values < 0.05.

## 3. Results

### 3.1. CD137 Signaling Plays a Protective Role in Systemic C. albicans Infection

To assess the function of CD137 signaling during systemic *C. albicans* infection, wild-type (WT) and *Tnfrsf9*^−/−^ C57BL/6 mice were infected with 3 × 10^5^ CFU *C. albicans* per mouse. *Tnfrsf9*^−/−^ mice displayed more rapid death than WT mice (Figure 1A), although the difference did not reach a statistical significance (*p* = 0.0528). These mice also experienced more severe loss of body weight than WT mice (Figure 1B). By contrast, injection of anti-CD137 agonistic antibody resulted in enhanced survival time after lethal challenge and slowed down loss of body weight (Figure 1C,D). There were significantly decreased levels of serum creatinine and BUN in anti-CD137 injected mice (Figure 1E). These results indicate that CD137 signaling protected the host from fatal renal injuries caused by *C. albicans* infection. Consistent with this interpretation, histopathological scores of the infected kidneys were significantly lower in anti-CD137-injected mice (Figure 1F).

### 3.2. CD137 Signaling Controls Fungal Growth and Renal Inflammation

Susceptibility or resistance to systemic *C. albicans* infection is determined by the host’s ability to repress growth of invaded fungi and lethal immunopathology [23]. Uncontrolled proliferation of pathogens is frequently linked to hyper-inflammatory responses that cause fatal tissue injuries. Accordingly, we found that anti-CD137-mediated resistance was associated with decreased outgrowth of *C. albicans* (Figure 2A). PAS staining of infected kidney sections revealed prominent hyphae within abscesses of control antibody-injected mouse kidneys, but this was not evident in anti-CD137-injected mouse kidneys (Figure 2B). This result suggests that anti-CD137 antibody effectively restricted fungi in the kidney.

Gross observations showed many more distinguishable nodules in control antibody-injected versus anti-CD137-injected mouse kidneys (Figure 3A). Consistent with this, histopathological analysis demonstrated that there were larger numbers of multifocal areas of abscess formation in control antibody-injected mouse kidneys (Figure 3B). These observations indicate that anti-CD137 antibody prevented extensive abscess formation by inhibiting fungal proliferation (also see Figure 2B). Furthermore, injection of anti-CD137 antibody slackened gain of kidney weight after infection (Figure 3C), an indication of less severe edema presumably as a result of lower degree of renal inflammation. Anti-CD137 antibody did not affect the weight of kidneys in uninfected mice (Figure 3C). Indeed, anti-CD137-injected mice had reduced levels of IL-6, TNF-α, CCL2, CXCL1, and CXCL2 in the infected kidneys (Figure 3D–F). There were also smaller numbers of infiltrating inflammatory monocytes and neutrophils in the kidneys of mice that received anti-CD137 antibody (Figure 3G). These results disclosed the association of anti-CD137-mediated reduction in fungal burden with lower levels of renal inflammation. However, as anti-CD137 antibody has been shown to protect the kidney from injuries induced by cisplatin or ischemia-reperfusion [21,24], the possibility that anti-CD137 antibody directly blocked renal inflammatory processes initiated by fungal infection is not excluded.

### 3.3. CD137 Signaling in Neutrophils Is Critical in Fungal Clearance

The results presented so far seem to suggest that CD137 signaling play a key role in controlling fungal clearance and fungal proliferation linked directly to renal inflammatory responses to systemic *C. albicans* infection. As neutrophils are indispensable for these two processes during an early phase of *C. albicans* infection, we first performed in vitro phagocytosis assays to examine whether or not CD137 signaling could directly act on neutrophils and enhance their phagocytic activity. We primed isolated neutrophils with anti-CD137 antibody for 1 h and then allowed primed neutrophil to phagocytize FITC-labeled, opsonized HK *C. albicans* for 20 min. Priming with anti-CD137 antibody significantly increased the percentages of neutrophils engulfing *C. albicans* (Figure 4B). There were higher levels of MFI in anti-CD137-primed neutrophils (Figure 4B). Since *Tnfrsf9*^−/−^ and WT neutrophils had a marginal difference in their phagocytic activity for *C. albicans* in in vitro assays (data not shown), anti-CD137 antibody can directly act on neutrophils to exert their priming effect. Pretreatment of U73122, a phospholipase C (PLC) inhibitor, abrogated the effect of anti-CD137 antibody on neutrophils’ phagocytic activities to a large extent (Figure 4B), indicating that CD137 signaling may activate the Dectin-1-CR3 phagocytic pathway in which PLCγ functions as a central signaling molecule [3,4]. We next performed in vivo phagocytocytic assays. After priming with anti-CD137 antibody for 1 h, FITC-labeled HK *C. albicans* were intraperitoneally challenged. Neutrophils primed with anti-CD137 antibody displayed increases in percentages of FITC-positive neutrophils as well as in individual neutrophils’ MFI (Figure 4C). Taken together, these results suggest that CD137 signaling not only broadened the pool of neutrophils with phagocytic capacity, but also increased the phagocytic activity of individual neutrophils.

Fungicidal assays demonstrated that phagocytosed *C. albicans* were killed more rapidly inside neutrophils primed by anti-CD137 antibody (Figure 4D), suggesting that priming with anti-CD137 antibody elevated the killing activity of neutrophils. Treatment of isolated neutrophils with *C. albicans* induced the production of higher levels of ROS after priming with anti-CD137 antibody (Figure 4E). Thus, CD137 signaling positively regulates a neutrophil’s capacity for killing by increasing ROS.

To evaluate the contribution of neutrophil-specific CD137 signaling to innate defense against systemic *C. albicans* infection in vivo, we transferred WT neutrophils into *Tnfrsf9*^−/−^ mice, followed immediately by injection of anti-CD137 antibody. Six hours later, mice were infected with a lethal dose of *C. albicans*. Pretreatment of anti-CD137 antibody significantly increased survival time and lowered fungal burden concomitantly (Figure 5A,B). When depleted of neutrophils in WT mice, the effect of anti-CD137 antibody on survival was completely abolished (Figure 5C), confirming that anti-CD137 antibody can increase survival by directly acting on neutrophils.

## 4. Discussion

Local innate immune responses to pathogens integrate an individual response of parenchymal and stromal cells, including tissue-resident and infiltrating immune cells, into an inter-connected cellular network to effectively manage infections [25,26,27]. Components in this network communicate with each other in two ways: soluble factors and their receptors and cell–cell interactions via membrane-bound ligand-receptor pairs. Cytokines and chemokines represent the former group and are of paramount importance in elimination of pathogens. Costimulatory and coinhibitory receptor pathways are representative of the latter group. They play a pivotal role in adaptive immunity to infections [28]. However, it is still to be clarified whether cell surface ligand-receptor interactions are involved in innate defense against invasive candidiasis. In this study, we demonstrate that CD137 expression on neutrophils is indispensable for fungal clearance. Our results indicate that priming of neutrophils by CD137 stimulation presumably with CD137 ligand (CD137L)-expressing cells seems to augment their capacity for phagocytosis and killing of *C. albicans*. As unprimed *Tnfrsf9*^−/−^ neutrophils have a marginal impairment in their phagocytotic capacity (data not shown), priming with CD137 signaling seems to occur during infection processes and contribute to elimination of invading *C. albicans*. However, it cannot be excluded that tonic signaling through CD137 in neutrophils primes them during differentiation processes in the bone marrow and during bloodstream circulation. The priming effect of CD137 signaling on phagocytosis was evident within 1 h either in vitro or in vivo (Figure 4A,B). CD137 expression solely on neutrophils was sufficient for lowering fungal burden in anti-CD137-primed mice (Figure 5A). It appears that CD137-induced PLC activation augments the phagocytic and killing activity of neutrophils (Figure 4D–F). The mechanism underlying these observations are currently unknown, but regulatory networks for the expression and activation of phagocytic receptors seem to be critical in this process [3,4,9,29]. Further studies will be guaranteed to clarify this aspect.

In the kidney, *C. albicans* infection induces release of IL-23 and IL-15 from dendritic cells and monocytes, respectively [22,30,31,32]. Activation of NK cells by these cytokines results in production of GM-CSF which is required for elimination of *C. albicans* by neutrophils [22,30,31,32]. Our results demonstrate that after *C. albicans* infection, tubular epithelial cells release a factor that induces IL-23 secretion by dendritic cells [Nguyen et al., unpublished data]. In addition, we have shown that CD137L/CD137 interactions between tubular epithelial cells and NK cells play a critical role in recruitment of neutrophils in ischemia-reperfusion kidney injury [21]. In aggregates, these results indicate that tubular epithelial cells, dendritic cells, monocytes, NK cells, and neutrophils are likely to form a cellular network responsible for renal inflammation whatever its causes are. Among these cells, CD137L is known to be expressed on tubular epithelial cells, dendritic cells, and monocytes/macrophages [21,33]. Therefore, these cells are candidates to stimulate neutrophils via CD137 in the kidney during *C. albicans* infection. Since CD137 is a costimulatory receptor for T cells and possibly for myeloid cells [34], CD137 signaling is likely to enforce primary signaling and thus function in a context-dependent way. In the case of neutrophils, the primary receptors seem to be a variety of pattern recognition receptors, immunoreceptors, and receptors for pro-inflammatory mediators [35,36,37]. Therefore, pathogens and inflammatory cells such as monocytes/macrophages, endothelial cells, and epithelial cells are likely to stimulate the primary signals through microbe-associated molecular patterns (PAMPs), danger-associated molecular patterns (DAMPs), and inflammatory cytokines [35,36,37]. As reverse signaling via CD137L triggers production of inflammatory mediators during inflammation [21,33,38], the CD137L-CD137 bidirectional signaling system may exert a positive feedforward regulation of neutrophils in fungal clearance [38]. This hypothesis is currently under investigation in our laboratory.

Hyper-inflammatory responses can result in fatal renal injury during invasive candidiasis without affecting resistance mechanism [39]. Our results indicate that CD137 signaling suppresses renal inflammation during systemic *C. albicans* infection by limiting fungal growth (Figure 3). This was caused at least partially by enhancement of neutrophils’ abilities of phagocytosis, killing, and growth restriction for *C. albicans*. However, a possibility cannot be excluded that potent stimulation of CD137 mobilizes regulatory types of immune cells such as dendritic cells, as seen in colitis [40]. Nonetheless, our results show that CD137 signaling in neutrophils is critical for innate defense against *C. albicans*. Manipulation of neutrophils with anti-CD137 agonistic antibodies could be considered as a therapeutic design for certain infections.

## Figures and Tables

**Figure 1 jof-07-00382-f001:**
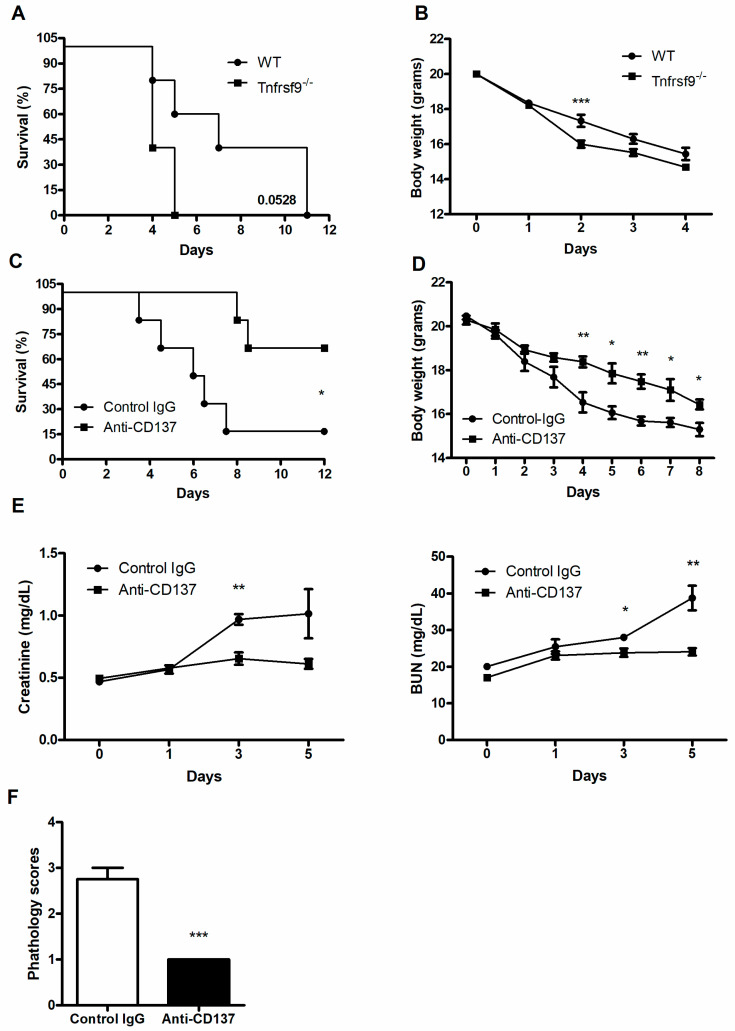
CD137 signaling protects the host from systemic *C. albicans* infection. (**A**–**D**) WT or *Tnfrsf9*^−/−^ C57BL/6 mice were infected with 3 × 10^5^ CFU *C. albicans*. (**A**) Survival rate (*n* = 5 mice per group). (**B**) Changes in body weight (*n* = 5 mice per group). (**C**,**D**) Mice were intraperitoneally injected with 200 μg of anti-CD137 or control rat IgG antibody 1 day before infection with 3 × 10^5^ CFU *C. albicans*. (**D**) Survival rate (*n* = 6 mice per group). (**D**) Changes in body weight (*n* = 6 mice per group). (**E**) Serum creatinine and BUN levels (*n* = 5 mice per group). (**F**) Histopathological scores of 3-day post-infection kidney sections (*n* = 5 mice per group). Results are representative of at least three experiments. * *p* < 0.05; ** *p* < 0.01; *** *p* < 0.001 between the two groups.

**Figure 2 jof-07-00382-f002:**
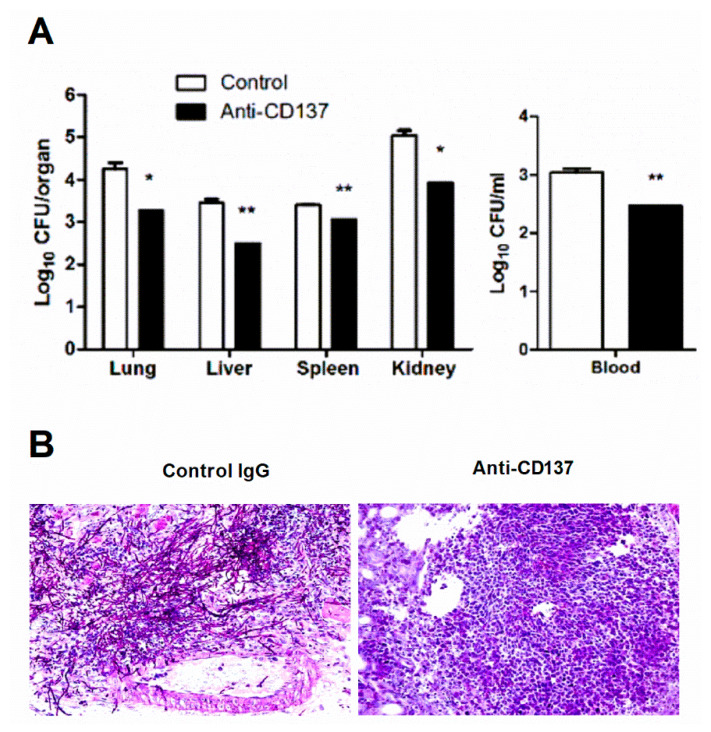
CD137 stimulation inhibits the outgrowth of *C. albicans*. (**A**,**B**) Mice were intraperitoneally injected with 200 μg of anti-CD137 or control rat IgG antibody 1 day before infection with 3 × 10^5^ CFU of *C. albicans*. Organs were harvested at 3-d post-infection. (**A**) Counting of CFUs (n = 5 mice per group). (**B**) PAS staining of kidney sections. Magnification: 200×. Results are representative of 3 experiments. * *p* < 0.05; ** *p* < 0.01 between the two groups.

**Figure 3 jof-07-00382-f003:**
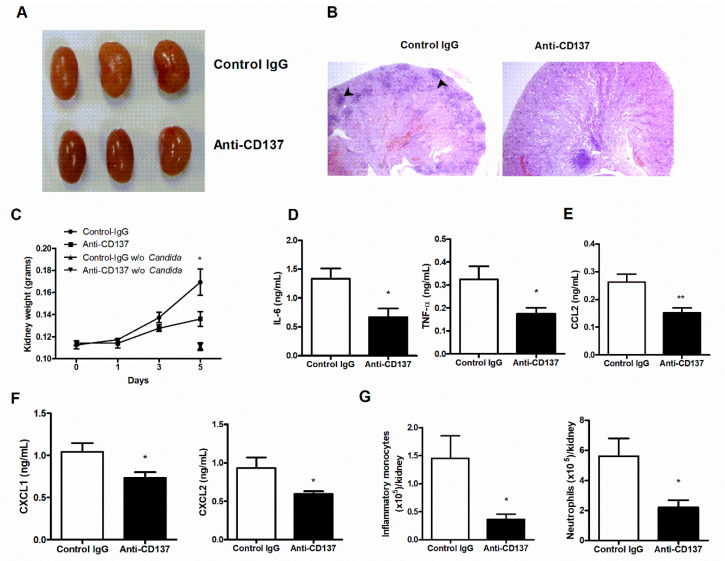
CD137 stimulation reduces renal inflammation after *C. albicans* infection. (**A**–**G**) C57BL/6 mice were injected intraperitoneally with 200 μg of anti-CD137 or control rat IgG antibody 1 day before infection with 3 × 10^5^ CFU of *C. albicans*. (**A**) Gross morphology. (**B**) Representative H&E staining of 3-d post-infection kidneys. Black arrowheads indicate representative nodules. Magnification: 40×. (**C**) Changes in kidney weight (*n* = 6 mice per group). In some experiments, mice were injected with 200 μg of control or anti-CD137 antibody without infection and kidney weight was measured 5 days after injection (*n* = 3 mice per group). (**D**–**F**) Levels of IL-6 and TNF-α (**D**), CCL2 (**E**), and CXCL1 and CXCL2 (**F**) in kidney lysates at 3-d post-infection (*n* = 5 mice per group). (**G**) Numbers of infiltrating inflammatory monocytes and neutrophils in 3-d post-infection kidneys (*n* = 5 per group). Results are representative of 3 experiments. * *p* < 0.05; ** *p* < 0.01 between the two groups at the indicated time points.

**Figure 4 jof-07-00382-f004:**
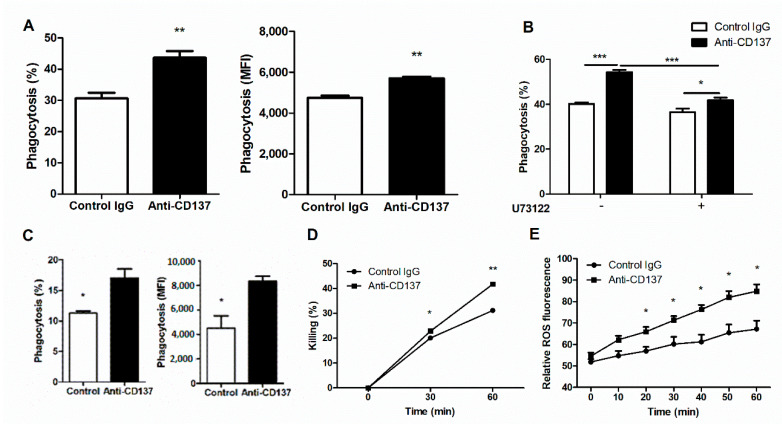
CD137 stimulation enhances the phagocytic and fungicidal activities of neutrophils. (**A**) In vitro analysis of phagocytosis. Purified neutrophils were preincubated with anti-CD137 mAb (5 µg/mL) for 1 h and challenged with opsonized, FITC-labeled HK *C. albiccans* (MOI = 10) for 20 min. The percentages and MFIs of FITC-positive neutrophils were presented for the extent of phagocytosis (*n* = 5 mice per group). (**B**) Neutrophils were pretreated with 2 μM U73122 (PLC inhibitor) 2 h before antibody treatment. The percentages of FITC-positive neutrophils were calculated using FACS. (**C**) In vivo analysis of phagocytosis. WT mice were intraperitoneally injected with 200 μg of anti-CD137 or control rat IgG antibody 1 h before challenge with FITC-labeled HK *C. albicans*. Cells harvested from the peritoneum were stained and percentages of phagocytosis were determined by counting CD11b^+^Ly6G^hi^ neutrophils containing *C. albicans*. The MFIs was also presented for the extent of phagocytosis (*n* = 5 mice per group). (**D**) Fungicidal assays for neutrophils primed with anti-CD137 or control antibody. (**E**) Measurement of ROS in neutrophils primed with anti-CD137 or control antibody. Results are representative of 3 experiments (*n* = 3 per group). * *p* < 0.05; ** *p* < 0.01; *** *p* < 0.001 between the two groups or between the indicated groups.

**Figure 5 jof-07-00382-f005:**
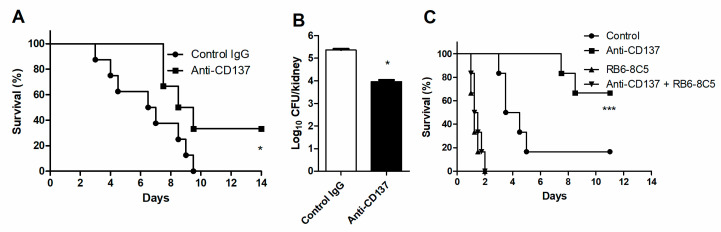
Neutrophils are required for anti-CD137-mediated enhancement of fungal clearance. (**A**,**B**) Neutrophils were isolated from WT mice and adoptively transferred to *Tnfrsf9*^−/−^ mice (5 × 10^6^ cells per mouse) immediately before injection with control or anti-CD137 antibody. Six hours later, mice were infected with 3 × 10^5^ CFU *C. albicans*. (**A**) Survival rates (*n* = 5 mice per group). (**B**) Fungal burdens in 3-d post-infection kidneys. (*n* = 5 mice per group). (**C**) Survival rates of mice depleted of neutrophils (*n* = 6 per group). Fungal burden in kidneys (*n* = 5 mice per group). Results are representative of 3 experiments. * *p* < 0.05; *** *p* < 0.001 between the two groups or between the indicated group and the other 3 groups.

## Data Availability

Not applicable.

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
