# Peer review of "CD137 Signaling Is Critical in Fungal Clearance during Systemic Candida albicans Infection"

_jof, 2021, doi:10.3390/jof7050382_

Round 1

Reviewer 1 Report

In this manuscript, Tran et al. describe the characterization of the surface receptor CD137, which has yet to be described in systemic candidiasis. They use knock out mice (Tnfrsf9-/-) and an anti-CD137 antibody agonist to explore the role of this surface receptor in vivo and in vitro. They conclude that CD137 signaling is protective against systemic C. albicans infection and functions to reduce fungal burden via increased neutrophil function, thereby reducing inflammatory organ damage. While the results generally support this conclusion, the methods lack important details required to fully interpret some of the data, and many experiments lack important controls, reducing the confidence in the study.

  1. The results displayed in Figure 1 A/C and B/D are misleading. The same inocula appear to be used in both sets of experiment, so why does the WT survival curve not match the Control IgG survival curve (A and C)? Why are different days used on the x-axes (A-D) and why are the weights reported in different units on the y-axes (B and D)? Additionally, an F panel is described in the results and the legend, however, there is no F panel in this figure.
  2. Due to the discrepancy between the WT and Control IgG survival curves described above, the inclusion of WT control groups throughout the entire manuscript should be strongly considered in addition to the Control IgG groups currently shown.
  3. The histopathological and gross morphology images (Figure 2B and 2A-B) are difficult to interpret; arrows should be added to aid the reader. Additionally, the quality of some of these images is very poor and should be enhanced.
  4. In Figure 3C, did you account for the additional weight caused by the higher fungal burden in the control kidneys? An uninfected control would be helpful to better interpret these results.
  5. Line 223: the data do no support that CD137 signaling “broadened the pool of neutrophils with phagocytic capacity”. As it is presented, the data do not differentiate between increased phagocytic capacity of individual neutrophils and increased numbers of neutrophils with phagocytic capacity. Exactly how the data presented in Figure 4A-B was calculated requires further description in order to clarify this issue.
  6. The U73122 inhibitor data is relatively weak and not enough to support the conclusion on lines 233-234. Furthermore, the wrong statistical comparison was carried out with relation to whether U73122 impacts CD137 signaling.
  7. Figure 5 has mismatched panels (A and B) and is missing critical information required to interpret the data, namely, whether the mice in each panel are WT or Tnfsf9-/-. Additionally, because the neutrophil depletion alone resulted in a high level of mortality, it is difficult to determine whether CD137 signaling has any role without knowing whether the mice use in this panel were WT or Tnfsf9-/- (or both).
  8. The figure legends throughout the manuscript require additional information, in particular, methods are not adequately described to fully understand/interpret the experimental designs.

Reviewer 2 Report

The objective of this study was to investigate the involvement of CD137 in the function of neutrophils in innate defense against systemic C. albicans infection. The proposal is innovative and the study was well designed with very interesting results. However, the subject and results were poorly explored in the introduction and discussion sections, respectively. In addition, various corrections are necessary in the text.  

Abstract:

Lines 10-12: “Neutrophils are particularly important for fungal clearance at the early phase of infections, yet little has been known regarding which surface receptor controls neutrophil phagocytic activities during systemic C. albican infection”. This sentence needs to be rewritten. The words “early phase of infections” and “during systemic infection” are confusing or repetitive.

Lines 14-15: Here, we used genetic and immunological tools to probe the involvement of CD137 signaling in innate defense mechanisms against systemic C. albicans infection. The word “neutrophils” must be included in the objectives.

Line 21: Replace “WT neutrophils” by “wild-type (WT) neutrophils. It is appears for the first time in the text.

Line 23: Delete this part from the conclusion of the abstract: “whereby harmful immunopathology-induced tissue injuries are minimalized”. There are no results in the abstract that can support this conclusion.

Introduction:

The introduction is poorly written. It needs a substantial revision based in the current literature. Just few old studies are mentioned. The introduction should be expanded with the inclusion of many other studies that bring new information to justify the proposal of the authors.

M&M:

-Lines 56-57: “C57BL/6 mice were purchased from Orient Bio-Charles River (Seongnam, Korea). Tnfrsf9-/- C57BL/6 [13] were maintained in a specific pathogen-free facility”. Describe how Tnfrsf9-/- C57BL/6 were obtained. Were they purchased or genetically modified in the laboratory? Specify the functional differences between C57BL/6 mice and “Tnfrsf9-/- C57BL/6”.

-Lines 76-86: The item 2.3 needs to be expanded. Describe in more detail the methodologies for “Isolation, depletion, and adoptive transfer of neutrophils. Maybe this item can be divided in 3 separate items.

-Line 87: Item 2.4 “Preparation of kidney cells and flow cytometry”. Correct this item because flow cytometry is described in the item 2.5.

-Lines 103-106 and 115-117: Describe in more details the items 2.6 and 2.8.

-Line 104: Describe the meaning of BUN?

Results:

-Line 176: The reference is missing: ‘immunopathology []”.

-Histological images (Figures 2B and Figures 3B) need to be replaced. It is not possible to identify the tissue structures and distinguished the cells.

Discussion:

-As well as introduction section, the discussion is poorly written and needs to be expanded with the inclusion of many other novel studies.

Round 2

Reviewer 1 Report

Tran et al. have done a nice job addressing my concerns and their additions have made their manuscript clearer. Although I did not request the changes to the intro and discussion, I feel that they have enhanced the work overall. I only have a few minor comments:

  1. While I appreciate and understand the issues with PDF conversion, I would still suggest that the authors consider adding arrows or other visual aids to help the reader identify the nodules (Fig. 3A) and multifocal areas of abscess formation (Fig. 3B) described in the text.
  2. The manuscript has a number of grammatical errors that need to be corrected. 

Author Response

We added arrowheads to indicate nodules in Figure 3B, as recommended. We also corrected English grammar errors in the manuscript.
